# Recent Advancements in mRNA Vaccines: From Target Selection to Delivery Systems

**DOI:** 10.3390/vaccines12080873

**Published:** 2024-08-01

**Authors:** Zhongyan Wu, Weilu Sun, Hailong Qi

**Affiliations:** 1Newish Biological R&D Center, Beijing 100101, China; wuzhongyan@newishes.com; 2Department of Life Sciences, Imperial College London, South Kensington Campus, London SW7 2AZ, UK; lillian.sun22@imperial.ac.uk

**Keywords:** mRNA, LNP, vaccines, virus, cancer, IVT

## Abstract

mRNA vaccines are leading a medical revolution. mRNA technologies utilize the host’s own cells as bio-factories to produce proteins that serve as antigens. This revolutionary approach circumvents the complicated processes involved in traditional vaccine production and empowers vaccines with the ability to respond to emerging or mutated infectious diseases rapidly. Additionally, the robust cellular immune response elicited by mRNA vaccines has shown significant promise in cancer treatment. However, the inherent instability of mRNA and the complexity of tumor immunity have limited its broader application. Although the emergence of pseudouridine and ionizable cationic lipid nanoparticles (LNPs) made the clinical application of mRNA possible, there remains substantial potential for further improvement of the immunogenicity of delivered antigens and preventive or therapeutic effects of mRNA technology. Here, we review the latest advancements in mRNA vaccines, including but not limited to target selection and delivery systems. This review offers a multifaceted perspective on this rapidly evolving field.

## 1. Introduction

Vaccines are one of the most effective measures for preventing and even eradicating infectious diseases. Through large-scale vaccination, humans completely eradicated smallpox in the 1980s, the deadliest infectious disease that had caused millions of deaths. Although polio and measles have not yet been completely eradicated, widespread vaccination has significantly reduced the incidence rate of these diseases, and the WHO is promoting the global eradication of polio, committed to eliminating more such diseases from the world [1].

Messenger RNA (mRNA) vaccines represent a revolutionary approach to infectious disease prevention and hold immense promise for cancer immunotherapy. Since the outbreak of COVID-19 and the success of two mRNA vaccines against SARS-COV-2, mRNA technology has quickly become the focus of vaccine research and development [2,3]. The advent of ionizable lipid nanoparticles (LNPs) revolutionized mRNA delivery, paving the way for their clinical application. Additionally, the development of modified nucleotides, such as pseudouridine, has further enhanced mRNA efficiency while minimizing immunogenicity, which has been applied to mRNA vaccines against COVID-19 [4,5]. mRNA vaccines induce the immune response to prevent and treat diseases by introducing genetic material encoding certain pathogen-specific antigens, which use the body’s self-manufacturing protein system to generate them instead of introducing the antigens to the body directly. This distinct strategy skips the complex pathogen cultivation and antigen extraction processes, greatly simplifying the production process and shortening the timeline of vaccines, which is crucial for a rapid response to emerging infectious diseases. mRNA vaccines have the following advantages compared to traditional vaccines: (1) safety—unlike DNA, mRNA vaccines avoid the risk of genomic integration; (2) high efficiency—mRNA vaccines can induce strong immune responses, including humoral and cellular immunity, providing highly effective protection [6]; (3) easy production—mRNA vaccine manufacturing is relatively simple as the production of mRNA does not require the use of live cells or viruses and uses no animal-derived raw materials and in vitro transcription (IVT) only takes a few hours; and (4) rapid development—the chemical structure of mRNA itself is independent of the antigen it encodes, and this characteristic endows mRNA vaccine production with modularity and flexibility. Therefore, mRNA vaccine production does not require adaptation when changing antigens. The quick design and manufacturing of mRNA vaccines allow rapid responses to emerging viruses and virus mutations [7,8].

Therapeutic mRNA vaccines represent a burgeoning class of immunotherapy distinct from traditional preventive vaccines. While traditional vaccines aim to pre-empt infection by stimulating the immune system, therapeutic vaccines focus on reprogramming the host’s immunity to combat existing diseases. By activating the adaptive immune system, these vaccines empower the body to recognize and eliminate established pathogens or abnormal cells, such as tumors. This therapeutic approach holds promise for chronic viral infections like HIV, HBV, and HPV [9,10,11]. In cancer treatment, therapeutic mRNA vaccines utilize tumor antigens to elicit potent anti-tumor immune responses, potentially controlling tumor growth, eliminating residual lesions, and establishing long-term immunological memory against cancer recurrence [12].

However, mRNA vaccines also face various challenges. Firstly, the large-scale nature of LNP formulation is costly and time-consuming. Encapsulating RNA in LNPs requires highly controlled mixing of RNA and lipids in the microfluidic chips, and it is hard to scale up to high-throughput production. Secondly, mRNA vaccines are highly sensitive to temperature, and their storage and transportation require ultra-cold-chain. Thirdly, mRNA vaccines may have strong side effects, such as fever and headache. The COVID-19 mRNA vaccine may lead to a T-cell-mediated autoimmune liver disease [13]. Moreover, the COVID-19 mRNA vaccine’s encoded spike protein is related to the risk of myocarditis, which is more pronounced in young males [14,15]. There is still significant room for improvements in mRNA vaccine development and application. This article will review the latest advancements in mRNA vaccines, particularly in sequence optimization, IVT, and delivery systems.

## 2. Results

### 2.1. Expansion of the mRNA Application

#### 2.1.1. The Application of mRNA in Infectious Diseases

The rapid response capabilities of mRNA vaccine technology were tested and verified during the COVID-19 pandemic. The SARS-CoV-2 mRNA vaccines promoted the germinal center responses and produced effective neutralizing antibodies, as well as antigen-specific polyfunctional CD4 and CD8 T cells [16,17]. Moreover, the second vaccination stimulated a notably enhanced innate immune response, which was concurrent with enhanced serum IFN-γ levels. Natural killer cells and CD8+ T cells in the draining lymph nodes are the major producers of this circulating IFN-γ [18]. Recently, Arcturus Therapeutics’ self-amplifying RNA (saRNA) vaccine against SARS-CoV-2 was approved in Japan, marking the world’s first saRNA vaccine [19]. saRNA is a type of mRNA molecule that has been engineered to replicate itself within host cells. saRNA utilizes the four non-structural proteins (NSP1-4) encoding the replicase of the alphavirus and the subgenomic promoter to achieve self-amplification, allowing for enhanced protein expression and a stronger immune response [20]. The application of mRNA vaccines has been widely extended to various diseases, including infectious diseases, rare genetic diseases, and immunotherapy in cancer [21,22]. Similarly, Moderna has a wide range of research and development pipelines, focusing on four sectors: respiratory viruses, latent viruses, cancer treatment, and rare disease treatment, showing the great potential and diverse applications of mRNA technology.

To date, mRNA vaccines have been applied in the research and development of multiple virus-related infectious diseases, including the influenza virus [23], RSV [24], norovirus, CMV [25], the Zika virus [26], VZV [NCT05701800], the rabies virus [27], and HIV [11,28]. In addition, mRNA vaccines show unique advantages for virus-related tumors, such as EBV, HBV, and HPV. Research on the HBV therapeutic mRNA vaccine indicates that mRNA vaccines targeting HBV can not only effectively eliminate the virus but also produce long-term protective antibodies, activate a strong immune response, and combat viral recurrence [29]. The saRNA vaccine against HPV-related tumors further expands the scope of treatment [30]. Our work showed that mRNA vaccines encoding HPV-E6E7 fused with chemokine CCL11 treatment achieved durable complete remission in tumor animal models [31].

Moreover, the potential of mRNA vaccine technology extends far beyond combating viral diseases, and its application boundaries are constantly expanding [32]. mRNA vaccines can also be applied to bacterial infections and parasitic diseases. saRNA encoding streptococcal antigens can effectively stimulate the immune response of mice to streptococci [33]. Similarly, in terms of malaria, saRNA Vaccines can induce specific immune responses against malaria parasites, and even achieve protection against reinfection through T-cell transfer in mice, opening up new avenues for the prevention and treatment of parasitic diseases [34].

Overall, mRNA technology is continuously driving the progress of vaccine science, not only accelerating the development of vaccines against current threats but also providing a flexible and efficient solution platform for possible future health challenges.

#### 2.1.2. The Application of mRNA in Cancer Therapy

Therapeutic vaccines, especially nucleic acid-based therapies, are leading a new era in cancer immunotherapy [35]. These vaccines not only treat common tumors with clear tumor antigens such as prostate cancer, non-small cell lung cancer, melanoma, multiple myeloma, breast cancer, and cervical cancer but also focus on personalized treatment by identifying and targeting tumor-specific neoantigens. For example, the BNT111 vaccine encodes four non-mutated tumor-associated antigens, which stimulate immune responses against multiple cancers. In phase 2 clinical trials, personalized vaccines such as BNT122 [36] and mRNA-4157 [37] have shown significant potential to prolong relapse-free survival, especially in inducing specific T-cell responses.

In addition, the flexibility of mRNA technology allows it to encode cytokines or monoclonal antibodies in tumor immunotherapy, such as IL12 mRNA combined with PD-1 antibody therapy [38]. Moreover, the direct intratumoral delivery of Moderna’s mRNA-2752 encoding OX40L/IL-23/IL-36γ produced robust anticancer responses in a broad range of tumor microenvironments [39]. On the other hand, mRNA technology is suitable for the production of antibodies in vivo, such as the BNT141-encoded CLDN18.2 monoclonal antibody, which aims to reshape the tumor microenvironment and directly target tumor cells [40]. Recently, BNT142 has been developed, which encodes bispecific antibody-targeting CD3 and CLDN6, and a phase 1/2 first-in-human clinical trial (NCT05262530) will be initiated [41].

Furthermore, mRNA-LNP technology is also used to directly deliver mRNA-encoded therapeutic proteins such as GSDMD [42] and TRAIL [43] to tumors, inducing tumor cell death and exploring new avenues for treatment. The innovative mRNA-LNP-CART technology represents a new direction for in vivo chimeric antigen receptor T cell therapy [44]. Up to now, CD3, CD4, and CD5 antibody-modified LNPs have been reported and verified for delivering mRNA into T lymphocytes in vivo [45].

Taken together, therapeutic mRNA vaccines and related mRNA technologies are a promising new approach to cancer immunotherapy.

#### 2.1.3. mRNA Technology in Other Diseases

mRNA technologies are leading a medical revolution, with their applications covering almost all disease areas. The in vivo production of therapeutic proteins becomes possible, including cytokines and monoclonal antibodies as mentioned before, bringing hope to patients with rare diseases and chronic metabolic diseases with in vivo protein replacement therapy. Researchers are exploring the potential of mRNA vaccines to treat a variety of rare genetic disorders, such as hereditary tyrosinemia type 1, phenylketonuria, methylmalonic acidemia, propionic acidemia, glycogen storage disease type 1a, and ornithine transcarbamylase deficiency [22]. mRNA-3927 encodes PCCA and PCCB subunit proteins to restore functional propionyl-coenzyme carboxylase activity in the liver. The positive results of mRNA-3927 [46] developed by Moderna in the clinical trial of treating propionic acidemia reduced the risk of life-threatening metabolic decompensation in patients by 70%, providing new hope for the treatment of rare genetic diseases and further demonstrating the broad prospects of mRNA therapy in the fields of precision medicine and rare disease treatment. The transgenic overexpression of the hepatocyte growth factor can increase insulin production and β cell proliferation [47]. Similarly, the transcription factors Pdx1 and MafA can reprogram pancreatic alpha cells into functional, insulin-producing β cells [48]. mRNA-LNPs targeting the pancreas hold immense potential for the treatment of type 1 diabetes by delivering mRNA encoding these proteins [49]. Moreover, mRNA technology has also been applied in genome editing by directly delivering Cas9 mRNA and gRNA simultaneously or separately [50].

Overall, mRNA technology provides innovative solutions for protein replacement therapy and gene editing and even shows potential application value in the field of beauty and anti-aging.

### 2.2. Antigen Selection and Design

#### 2.2.1. Antigen Selection: The Key to Success in Vaccine Development

The selection of antigens is crucial for the success of vaccines, as it requires a profound understanding of the complete life history of the pathogenic microorganisms that cause the corresponding diseases, as well as their underlying mechanisms. For tumors, it is necessary to identify tumor-specific or highly expressed antigens. For example, the E6 and E7 proteins of HPV are used to prepare therapeutic vaccines due to their latent expression after viral invasion, while the outer shell proteins L1 and L2, which are responsible for the interaction between HPV and host cells to assist virus invasion, are used to prepare preventive vaccines [10].

After selecting the antigen, it is essential to conduct a detailed investigation and analysis of its biological function. On one hand, it is necessary to avoid the risk of overexpressing the natural oncogenic antigen. For instance, the HPV E6 protein has the ability to degrade P53 and is considered a proto-oncoprotein; thus, mutations in the corresponding binding sites are required in vaccine design [51]. On the other hand, the appropriate conformation helps for the effective immune response. In the design of COVID-19 vaccines, a double proline mutation in the S protein is used to stabilize its pre-fusion conformation [52]. This aims to make the expressed S protein more stable in the conformation before binding to host cell receptors, thereby producing neutralizing antibodies that block its invasion into host cells.

The selection of antigens has been extensively reviewed elsewhere, and this article will focus on the recently applied epitope optimization strategies.

#### 2.2.2. Epitope Optimization Enhances the Adaptive Immune Responses

An antigenic epitope is the basic unit that triggers cellular or humoral immune responses. The optimization of vaccine antigen epitopes is another key step in vaccine design, aiming to enhance immunogenicity and protective efficacy, break immune tolerance, and provide cross-protection for multiple variants through the selection and modification of antigenic determinants. This process requires a deep understanding of the immunological characteristics of pathogens and the use of advanced biotechnology and computational tools. There are several specific optimization strategies involved. (1) The first is epitope selection, which concerns identifying conserved epitopes that can induce broad protective immunity and are not easily mutated by pathogens, utilizing linear epitope mapping analysis to guide the selection of epitopes closely related to disease severity, and neutralizing activity, which ensure the efficiency and durability of vaccines [53]. For pathogens such as HIV or certain tumor antigens that can evade the host immune system through mutated epitopes, vaccines that target hidden or less mutated epitopes should be developed. Optimizing vaccine immunogens based on consensus sequences of viral evolution helps to develop broad-spectrum vaccines to combat future mutations [54]. (2) The second is multi-epitope design. Multi-epitope vaccines constitute a promising strategy against tumors and viral infections. Compared to classical vaccines and single-epitope vaccines, multi-epitope design aims to enhance the immunogenicity of vaccines and cover immune responses in different populations. Multi-epitope design includes (I) multiple B or T epitopes, (II) combined B and T cell epitopes, and (III) multiple epitopes from different tumor or virus antigens [55]. The monkeypox mRNA vaccine MPXVac-097 combined five MPXV antigens that have been identified as targets of neutralizing antibodies that induce broad-spectrum neutralizing antibodies and specific T cell responses by tandemly linking five antigens on a single mRNA molecule, providing protection against live virus challenge in animal models [56]. Multi-epitope vaccines consisting of B and T cell epitopes have the potential to trigger innate, adaptive, and humoral responses [57]. mRNA vaccines encoding HLA-EPs (T epitopes) and the receptor-binding domain of the SARS-CoV-2 (neutralizing antibodies) produce optimal protection against SARS-CoV-2 in nonhuman primates [58]. Multivalent HPV DNA/mRNA vaccines encoding HPV16/18 E6/E7 epitopes induced significant antigen-specific cellular immune responses in mice CIN3 patients [59,60]. (3) The third strategy is structure optimization. By utilizing molecular biology and computational biology methods, the structure of epitopes can be adjusted to enhance their binding ability with immune cell receptors or stabilize the epitope’s conformation to better mimic antigens under natural infection conditions [52,61]. Lastly, (4) there is AI-assisted design. AI technology can be used to accelerate the discovery and optimization of antigen epitopes, predict the immunoreactivity of epitopes, and optimize vaccine design, especially when dealing with complex datasets and predicting multi-dimensional responses during interactions, demonstrating significant advantages [62].

Through these advanced epitope optimization methods, we can design more efficient, safer, and broadly protective vaccines against rapidly mutating viruses.

### 2.3. mRNA Sequence Optimization

#### 2.3.1. Codon Optimization Contributes to Antigen Expression

The codon contents influence the translation and stability of mRNA [63]. Codon optimization is a pivotal strategy employed to enhance the protein expression of mRNA. The optimization encompasses several strategic adjustments: (1) optimal codon: codon usage bias and GC nucleotide content are two common tools for optimizing protein expression; (2) secondary structure: the three-dimensional folding pattern formed by mRNA is a secondary structure, and optimization aims to lengthen the mRNA half-life and minimize unfavorable structures that impede translation; (3) translational regulation: achieving efficient protein expression requires a balance between the translation initiation speed and the fluency of the subsequent translation process. Furthermore, managing the ribosome load on the coding sequence (CDS) is instrumental in prolonging mRNA stability and boosting protein output. High ribosome load on CDS accelerates mRNA decay, limiting protein synthesis over time, highlighting the importance of load optimization in mRNA design for enhanced expression outcomes [64]. Notably, the Linear Design Algorithm, introduced by Zhang et al., innovatively integrates the Codon Adaptation Index (CAI) and Minimum Free Energy (MFE) considerations to engineer more stable mRNA sequences, augmenting intracellular protein synthesis [65].

#### 2.3.2. The Shadowed UTRs Determine the Fate of mRNA

The 5’ and 3’ untranslated regions (UTRs) of mRNA, which are less likely to attract attention compared to the coding sequence, play a critical role in protein expression. These flanking regions function as a complex regulatory unit, influencing translation efficiency, intracellular localization, and mRNA stability through diverse mechanisms. The 5’ UTR of mRNA can form different secondary structures, and the secondary structure affects ribosome loading and translation initiation, thereby regulating translation efficiency [66]. To enhance overall mRNA expression, researchers refer to the 5’ UTR sequences of highly expressed genes, like β-globin. Similarly, studies have found that the 3’ UTR sequence of specific genes (such as hα-globin, FAM171A1, C3, TIAM1, and POTEE) can significantly improve the expression level of mRNA. The 3’ UTR not only affects the stability and degradation rate of mRNA but also regulates the localization of mRNA in the cytoplasm, which in turn affects the expression level and type of proteins [67]. By engineering the design of the 3’ UTR, the distribution of mRNA can be altered, providing strategies for enhancing protein expression. The optimization of UTRs in CureVac AG’s second-generation mRNA vaccine CV2CoV improved the immunogenicity and protective efficacy of the mRNA SARS-CoV-2 vaccine [68]. The UTR of CV2CoV supports higher protein expression, which is composed of the 5′ UTR from the human *HSD17B4* gene and a 3′ UTR from the human *PSMB3* gene [69]. The advent of deep learning models has revolutionized RNA design, such as RNAdegformer [70] and UTR-LM [71]. These models use deep learning to predict RNA degradation rates and regulatory functions, fostering a deeper understanding of mRNA regulatory mechanisms. By successfully predicting and experimentally validating novel strategies to enhance protein synthesis, these models hold immense potential for future research.

Additionally, the inclusion of microRNA binding sites such as miR122 in the 3’ UTR, which is highly expressed in hepatocytes, helps to attenuate potential protein expression in hepatocytes, ultimately alleviating hepatotoxicity [39]. Moreover, the miRNA response elements can turn on/off the expression of mRNA [72]. A recent study presents an engineered 3’ UTR composed of an RBP-specific aptamer region and an shRNA- or HHR-mediated cleavage site, which is helpful for translational regulation in mammalian cells [73].

In summary, by precisely designing and optimizing the non-translated region of mRNA, especially using bioinformatics tools and deep learning models to predict its structure and function, we can effectively regulate the translation efficiency, cellular localization, and stability of mRNA, opening up new possibilities for mRNA-related therapies.

#### 2.3.3. The Long and Structured Poly(A) Tail in mRNA Vaccines

The 3’ poly(A) tail plays a key role in mRNA destiny. A longer poly(A) tail can protect mRNA from deadenylation and degradation, ensuring mRNA stability and efficient translation. Both the length and structure of poly(A) are crucial for mRNA’s stability and translation rate [74]. Approved mRNA vaccines provide a typical example of the structure of the poly(A) tail. The BNT162b2 employs a unique poly(A) sequence containing a specific linker GCAUAUGACU (30A + 10GCAUAUGACU + 70A), while the mRNA-1273 utilizes a simpler, homogenous poly(A) tail consisting solely of 100 adenines. The use of segmented poly(A) can significantly reduce plasmid recombination in *E*. *coli* without any negative effects on mRNA half-life and protein expression [75]. A study using systematic substitution of non-A nucleotides showed that cytosine (C) within the 3’ poly(A) tail effectively enhances the protein expression and duration of synthetic mRNAs both in vitro and in vivo [76]. The introduction of cytosine prevents the mRNAs from the deadenylation of the CCR4-NOT transcription complex. In addition, the structure of the poly(A) tail affects the stability and translation capacity of mRNA. The multiple synthetic topologically and chemically modified mRNA poly(A) tails enhanced the translation capacity of mRNA [77].

There are two main strategies for the in vitro synthesis of poly(A) tails (Figure 1): one is enzymatic synthesis, which uses poly(A) polymerase to synthesize poly(A) tails after mRNA transcription is completed, and the other is co-transcriptional synthesis, where the poly(A) tail is directly transcribed from a poly(A) sequence already present on the template plasmid DNA used for IVT. Moreover, only co-transcriptional synthesis can generate segmented poly(A). A recent study using click chemistry to stabilize poly(A) provides more options for customized production of mRNA [77].

### 2.4. In Vitro Transcription

#### 2.4.1. Modified Nucleotides Protect mRNA from Immune Systems

The modification of RNA plays a crucial role in guiding the immune system. Specific modifications of RNA (such as m5C, m6A, m7G, inosine, and 2’-O-methylation) assist the immune system in distinguishing host RNA and foreign pathogenic RNA and regulating immune responses [4]. Unmodified mRNA inherently possesses immunogenicity due to its interaction with Toll-like receptors, particularly in antigen-presenting cells like dendritic cells (DCs) and plasmacytoid dendritic cells (pDCs). Modified nucleotides such as m5C, m6A, m5U, s2U, or pseudouridine can effectively reduce the immune stimulation of mRNA. Katalin Kariko’s pioneering work highlighted that mRNA modified with N1-methyl pseudouridine (N1mΨ-UTP) exhibits the highest translation efficiency, and this finding has been applied to SARS-CoV-2 mRNA vaccines to significantly improve their performance [5]. Modified nucleotides offer a multifaceted advantage, optimizing mRNA vaccine stability, effectiveness, and safety. They can enhance translation efficiency, reduce immunogenicity, and ultimately contribute to a safer vaccine profile. A compelling example lies in CureVac’s CVnCoV vaccine, which utilizes unmodified nucleotides. This approach resulted in a lower observed protective efficacy (48.2%) compared to vaccines like BNT162b2 and mRNA-1273 that use modified nucleosides [78].

Recent evidence indicates that the saRNA modified by N1mΨ-UTP has limited protein expression in cells [79], which may be due to the inhibition of NSP4 (RNA-dependent RNA polymerase, RdRp), which cannot efficiently identify modified mRNA for replication. In addition, an unexpected finding is that after vaccination with BNT162b2, nonspecific T cell responses to +1 frameshift mutant antigens were observed, which is related to the translation frameshift triggered by N1mΨ-UTP [80]. The incorporation of 5 methylcytidine alleviates the innate immune response to saRNA vaccine [79]. This challenges the prior assumption that modified nucleosides might hinder saRNA vaccines. VLPCOV-02, a SARS-CoV-2 saRNA vaccine with a modified 5-methylcytosine, reduced the reactogenicity of saRNA and exhibited good safety and efficacy in Phase I clinical trials [81]. The ability of 5mC to modulate the innate immune response opens exciting new avenues for future vaccine design. Moreover, 5-OH-mC and 5mU also show enhanced expression and Reduced interferon response [82].

In conclusion, modified nucleotides play a critical role in fine-tuning the delicate dance between mRNA vaccines and the immune system (Figure 2). As research delves deeper, a nuanced understanding of these modifications will continue to shape the landscape of safe and efficacious mRNA vaccines.

#### 2.4.2. Cap Analogs Empower Co-Transcriptional Capping

The 5′ m7G cap is a highly conserved structure on eukaryotic mRNA, which recruits essential proteins, mediating functions like pre-mRNA processing, nuclear export, and ultimately, protein synthesis through a cap-dependent mechanism [83]. Cap-0 is a N7-methyl guanosine connected to the 5′ nucleotide through a 5′ to 5′ triphosphate linkage. The Cap-0 structure is essential for the efficient translation of the mRNA. An additional methylation on the 2′O position of the initiating nucleotide generates Cap-1. Cap-2 is similar to the Cap-1 structure, but it also has an additional methylation in the 2′O position of the second nucleotide. Despite its low cellular abundance, the Cap-2 structure significantly impacts mRNA stability and translation efficiency. The Cap-2 structure is crucial for reducing the activation of the innate immune response [84]. Notably, viruses like SARS-CoV-2 encode enzymes (2′-O-MTase) to strategically modify their RNA caps, facilitating immune evasion and promoting stable translation of viral RNA [85].

In the manufacturing of mRNA vaccines, the cap structure is crucial. Typically, Cap-1 structures are introduced through post-transcriptional capping or co-transcriptional methods using cap analogs. Among these, Anti-Reverse Cap Analogs (ARCA) were initially favored due to their positive capping properties [86], although early capping efficiency issues limited their widespread application. Moreover, additional enzymatic reaction steps are required to convert Cap0 into Cap1 structure. However, with the emergence of a new generation of Cap1 analogs, CleanCap provides a highly efficient and low immunogenic capping solution, significantly improving the performance of mRNA vaccines [87]. During IVT, CleanCap does not compete with GTP, which ensures superior capping efficiency. Overall, cap analogs serve as an essential component of mRNA vaccines, improving translation efficiency, immunogenicity, and safety profiles. With further research, cap analogs are expected to play a greater role in the development and application of mRNA vaccines [88].

Beyond the classic cap analogs, researchers are exploring specialized cap analogs with additional functionalities. Specially designed cap analogs can also promote the purification of mRNA and enhance its translation ability in vitro and in vivo, such as hydrophobic tags [89] and benzylated capAnalog m7GpppBn6AmpG [90]. In addition, photosensitive caps like FlashCaps enable spatial and temporal control of mRNA expression, further expanding the therapeutic potential of mRNA technologies [91].

The combined modification of N1mΨ-UTP and Cap-1 effectively balances inflammatory responses while enhancing adaptive immunity, particularly in influenza vaccine models. Compared to unmodified or single-modified mRNAs, these dual-modified vaccines exhibit superior gene expression profiles and protective effects, highlighting the importance of the precise regulation of the mRNA structure for optimizing vaccine performance [92].

#### 2.4.3. Multiple Measures to Minimize the Production of dsRNA

Double-stranded RNA (dsRNA), a by-product of IVT generated by T7 RNA polymerase (T7 RNP), presents a double-edged sword in mRNA vaccine development. Several primary mechanisms contribute to dsRNA formation during IVT with T7 RNP, including RdRp activity, in which T7 RNP’s inherent RdRp activity can lead to self-pairing of the transcribed RNA, forming hairpin structures. Additionally, incomplete transcripts might pair with each other, generating double-stranded regions. Additionally, there is off-target transcription, whereby, in rare instances, T7 RNP can transcribe the DNA template strand using an independent promoter, leading to the formation of dsRNA.

Despite dsRNA’s potential adjuvant effect in mRNA vaccines [93], it is considered a potential source of side effects of RNA vaccines due to the stimulation of the immune response. To reduce the content of dsRNA in mRNA drug substances, general strategies for optimizing IVT conditions including reducing magnesium ion concentration, incorporating modified nucleotides, and utilizing mutated enzymes have been applied. Athanasios Dousis et al. designed a double-mutant T7 RNP, which produced significantly less immune-stimulating RNA in IVT compared to the wild type [94]. Moreover, the amount of dsRNA generated is affected by the quality of linear plasmid templates [95]. Additionally, utilizing urea as a separation agent can significantly inhibit the formation of dsRNA [96]. Post-transcriptional processing uses techniques like phenol-chloroform extraction or column chromatography to remove reaction components and contaminants. Purification by cellulose can remove 90% of the dsRNA contaminants, offering an effective, reliable, and safe method [97]. By employing a multifaceted approach that combines optimized IVT conditions, advanced purification techniques, high-quality templates, and even engineered polymerases, dsRNA content has been effectively minimized in mRNA vaccines.

### 2.5. Delivery Systems for mRNA Vaccines

Although early studies showed that naked mRNA can be directly injected without vectors to express the protein in mouse muscle cells [98], the inherent instability of RNA molecules, susceptibility to RNase degradation, and negatively charged nature have limited their cellular uptake and clinical potential. Lipid-based delivery systems and Polymer-based delivery systems are two important mRNA delivery platforms. With the emergence of efficient mRNA delivery vectors, especially the success of the COVID-19 vaccine delivery system, LNPs containing ionizable cationic lipids have revolutionized mRNA therapy.

#### 2.5.1. Lipid-Based mRNA Delivery Systems

Lipid-based delivery systems, such as liposomes, lipoplexes (LPXs), LNPs, cationic nanoemulsions (CNEs), SLNs, and NLCs, are the most advanced mRNA delivery systems in the clinic (Figure 3).

A liposome is a type of small vesicle with a structure similar to the cell membrane, composed of a closed bilayer phospholipid, and is the most classic lipid-based delivery system [99]. Liposome has advantages such as low toxicity and high biocompatibility, with low encapsulation efficiency. In order to improve the interaction with LPs, mRNA can be formulated with cationic LPs to form LPXs. Cationic lipids, which have a positive surface charge, interact electrostatically with the negatively charged phosphate groups of the DNA or RNA, leading to the formation of stable aggregates or particles. DOTMA and DOTAP have been used for mRNA delivery alone or in combination with other materials. Early cationic lipids such as DOTAP effectively load nucleic acids, but their application is limited due to toxicity in vivo [100].

The introduction of ionizable cationic lipids such as MC3 has achieved low toxicity and high-efficiency delivery. LNPs utilize the unique properties of ionizable lipids [101]. At low pH values, when the lipids are protonated and positively charged, they are efficiently complexed with negatively charged nucleic acids. As the pH rises to physiological levels, the loaded nucleic acids are retained within the LNP. Following cellular uptake, the acidic environment within endosomes triggers the re-protonation of the ionizable lipids. This facilitates endosomal escape through the formation of ion pairs with endosomal phospholipids, ultimately promoting the release of the therapeutic cargo. In contrast to liposomes, LNPs exhibit a more diverse structural repertoire, featuring a multilayered core composed of alternating lipid and nucleic acid rings. Lipid nanoparticles containing siRNA synthesized through microfluidic mixing exhibit electron-dense nanostructure cores [102]. Cryo-TEM and X-rays show that siRNA and ionizable lipids form tightly packed bilayer complexes at pH 4, where siRNA is sandwiched between closely apposed monolayers. At physiological pH, these small vesicular structures coalesce to form larger LNP structures with amorphous electron-dense cores [103]. A further method based on the multi-laser cylindrical illumination confocal spectroscopy technique quantified the Payload distribution of mRNA-LNP, a commonly referenced benchmark formulation using DLin-MC3 as the ionizable lipid mostly contains two mRNAs per loaded LNP with a presence of 40–80% empty LNPs [104]. ApoE was once considered to be the key to LNP liver targeting [105], but new research suggests that HDL may play a more important role [106]. The year 2018 witnessed a landmark achievement with the FDA approval of the first siRNA drug delivery system based on an ionizable lipid formulation, which marked the recognition of the reliability of the RNA drug delivery strategy using LNP. Subsequently, during the COVID-19 pandemic, mRNA-LNP vaccines emerged as a beacon of hope in the fight against the virus. The successful application of these vaccines further promoted the research and development of LNP. Structurally, ionizable lipids can be divided into various types such as unsaturated, multi-tailed, polymer, degradable, and branched tails, and continuous optimization is needed to address current limitations [107].

Lipopolyplex (LPP) technology combines the advantages of polyplexes and LPXs. It offers excellent colloidal stability, reduced toxicity, and enhanced gene transfer efficiency [108]. The COVID-19 vaccine SW0123 from Stemirna Therapeutics adopts this delivery strategy [109], and the clinical trial application for the LPP-based mRNA vaccine has been accepted by the CDE (China’s National Medical Products Administration) and clinical trials have been conducted. In addition, the LPP-based mRNA vaccine has been granted Emergency Use Authorization by Laos.

#### 2.5.2. Lipids for Advanced LNPs

To address the challenges of mRNA-LNP delivery efficiency and extrahepatic targeting, two primary strategies have emerged: the development of novel ionizable cationic lipids and the incorporation of a fifth component into the existing quadruple-component system.

Chemical bonds at the tail of ionizable cationic lipids influence the targeting of LNPs. For instance, O-series LNPs (containing an ester bond in the tail) tend to deliver mRNA to the liver, while the N-series LNPs (containing an amide bond in the tail) to the lung [110]. High-throughput screening facilitates the delivery of mRNA to specific organs. In addition, the innovative design enables LNP to various cell-type targeting. The delivery of mRNA to pancreatic beta cells [49] or the promotion of sperm production through LNPs constructed from CAP lipids [111], as well as specific delivery to placenta [112,113], microglia [114], and spermatocytes, show potential for application in various biological scenarios.

LNPs have revolutionized mRNA delivery but conventional LNPs exhibit preferential accumulation in the liver. Selective Organ Targeting (SORT) technology addresses this limitation by incorporating a fifth, custom lipid into the LNP, enabling precise delivery to various target organs, including the lungs and spleen [115]. This approach achieves specific splenic T cell targeting [116] and bone marrow targeting [117] delivery, and the SORT LNPs are also applicable for CRISPR-Cas gene editing applications [118]. Moreover, the utilization of the SORT methodology can enhance mRNA expression by adding efficient endosomal escape lipids, which are called Syn-LNPs [119].

Moreover, to simplify the non-liver mRNA delivery systems, one-component ionizable cationic lipids rich in secondary amines have been designed to efficiently deliver mRNA to the T cell subsets following intravenous administration [120]. Non-cationic thiourea lipid nanoparticles (NC-TNPs) encapsulate mRNA using unique hydrogen bonding interactions, avoiding traditional charge attraction, simplifying the preparation process, reducing side effects, and demonstrating efficient transfection and spleen-targeting capabilities [121]. Multilayer RNA–lipid particle aggregates (LPAs) represent another advancement, particularly for tumor immunotherapy [122].

Nebulized formulations of mRNA-loaded LNPs constructed using P76 (a poly-β-amino thioester polymer) [123] or hPBAE [124] offer a promising approach for inhaled mRNA delivery, significantly improving lung mRNA delivery. Advances in stability include the development of responsive lipid molecules that can maintain the stability of mRNA while ensuring effective release [125], as well as ionizable lipids derived from soybean oil that simplify the synthesis process while ensuring efficient and safe mRNA delivery performance [126]. These technological advances have collectively promoted the diversification and efficiency of mRNA delivery strategies.

#### 2.5.3. Polymer-Based mRNA Delivery Systems

The polymer-based delivery system, also known as polyplex, formed by the electrostatic interactions between cationic polymers and negatively charged nucleic acids offers a protective shield for mRNA. The protective complexes protect nucleic acids from degradation by nucleases, enhancing the stability and efficiency of nucleic acid delivery. While polymer-based mRNA delivery systems show lower transfection efficiency and potential toxicities than lipids, they have the potential for the assembly of various nanostructures in aqueous conditions, lyophilization, and long-term storage [127]. The polymers used in Polyplexes mainly include PEI, polyesters, poly(amino acids), and dendritic polymers.

PEI is a cationic polymer and has long been utilized for nucleic acid delivery, forming complexes through electrostatic interactions. However, their non-degradability and relatively poor biocompatibility limited their clinical application.

Polyesters are excellent in biocompatibility, biodegradability, and biosafety. A variety of polyesters have been used for mRNA delivery, such as PLGA, PBAEs, APEs, and PACE. The application of degradable polymers PLGA and PACE has shown potential for controlled release and efficient transfection [128,129]. In particular, hyperbranched poly(β-amino ester) (hPBAE) [130] helps the inhalation delivery of mRNA to the lungs, exhibiting high efficiency and low toxicity [123].

Poly(amino acids) like poly(lysine) have also long been utilized for nucleic acid delivery. Poly(amino acids) contain amphiphilic block copolymers and, thus, can generate a specialized core–shell structure. A recent study used poly(l-ornithine) and a Charge-Conversion Polymer to protect mRNA and improve protein expression efficiency [131].

Dendrimers are a class of highly ordered branched polymer molecules with broad internal cavity structures and dense surface-active functional groups. Polyamidoamine (PAMAM) is one of the most extensively studied dendrimers. pABOL is a bioreducible, linear, cationic pAMAM, which is optimal for saRNA [132]. The constructed mRNA nanoparticle liposomes successfully achieved lung-targeted delivery. pABOL formulations resulted in 100-fold higher intramuscular protein expression than LNP [133].

#### 2.5.4. Other Promising mRNA Delivery Systems

Peptide delivery technology has also been explored in the field of mRNA delivery. Protamine, as an attempt, can enhance the transfection efficiency of the CV7201 [101] rabies vaccine, but it was ultimately abandoned due to toxic side effects. CPPs are suitable for peptide-based mRNA delivery. Cell-penetrating peptides (CPPs) are peptides with cell membrane permeability. In general, CPPs are internalized by cells via direct cell membrane penetration and/or endocytosis [134].

Virus-like particle (VLP) delivery has become a hot topic, especially in CRISPR/Cas9 mRNA therapy research, which shows potential. Gag and the mammalian Gag homologs PEG10 that recognize the stem loop of mRNA have been used to package, secrete, and deliver specific RNAs [135,136]. eVLP is a DNA-free engineered VLP [137], and using different glycoproteins in eVLPs achieved the targeting of VLPs [138].

Another promising direction concerns Extracellular vesicles (EVs) and extracellular secretion. As an emerging vector, the body shows the prospect of delivering mRNA. IL-12-Exo promotes immune activation and memory in the treatment of lung cancer with exogenous mRNA [139], while engineered EV can effectively deliver collagen mRNA to the skin to combat aging-related diseases [140]. In addition, exogenous mRNA can be delivered to the skin through engineered EV to promote the production of collagen and reduce wrinkles. The body delivery strategy targets neurons [141], overcomes the blood–brain barrier, and provides a new approach to the treatment of brain diseases. Similarly, the bionic red blood cell membrane encapsulation technology reduces immune response and improves delivery safety [142].

In terms of microbial delivery, yeast and bacteria also show potential as delivery vectors [143,144], further expanding the implementation methods of mRNA therapy.

In summary, mRNA delivery technology is constantly advancing, and polymer and lipid-based delivery systems each have their own characteristics. New materials and technologies such as VLP, exosomes, and gene-modified viruses are also being explored. In the future, the optimization of LNP delivery for specific tissues will be a key direction of research and development.

#### 2.5.5. Surface Modification Expands the Applications of LNPs

Surface modifications improve the targeting ability of LNP functions. The most direct and effective strategy is the conjugation of antibodies to LNPs, which expands the applications of LNPs. In vivo CART refers to the production of CART cells in vivo, without the need for ex vivo modification and transfusion. By coupling CD5 antibodies on the surface of LNPs (Ab-LNPs), these particles are used to carry CAR mRNA and target T cells for In vivo CART production [44]. This platform utilizes Ab-LNPs [145] to achieve precise delivery, induce temporary and controllable expression of CAR, effectively reduce non-target effects and adverse reactions, and promote the safety and controllability of CAR-T therapy.

The mRNA vaccine co-modified with SA-AE-AC-CH sialic acid cholesterol derivative and breakable PEG lipid achieves targeted delivery to DCs and accelerated endosomal escape, greatly enhancing the expression of antigen proteins in DCs, optimizing the efficacy of mRNA vaccines, and reducing side effects [146]. Similarly, Ligand-tethered LNPs represent a promising strategy for targeted RNA delivery, offering enhanced efficacy and reduced off-target effects compared to conventional LNPs. This targeted approach holds immense potential for treating liver fibrosis, pulmonary diseases, and other disorders by selectively delivering RNA payloads to specific cell types [147]. Another promising strategy concerns anisamide-tethered lipidoids (AA-lipidoids). Anisamide is a high-affinity ligand for the sigma receptor that is highly expressed on rapidly proliferating activated fibroblasts, including activated HSCs. Other motifs, including aptamers, can also be incorporated for targeting by coupling to hydrophilic PEGylated lipids.

### 2.6. Adjuvants

Adjuvants are ingredients added to some vaccines to amplify the immune response to the vaccine. Several well-established adjuvants are routinely employed to bolster the immune response, including alum salts, Toll-Like receptor (TLR) agonists, and oil-in-water emulsions. These adjuvant strategies have jointly promoted the innovation of vaccine technology and improved the efficiency and specificity of immune protection: (1) Aluminum adjuvants enhance the immune response to vaccines by forming antigen depots and stimulating immune cells. (2) Pathogen-associated molecular patterns directly stimulate TLRs expressed on antigen-presenting cells (APCs). (3) Oil-in-water emulsions, exemplified by MF59 and Squalene, act as delivery vehicles for vaccine antigens. They encapsulate the antigens, facilitating their uptake by APCs. Adjuvants are classified as immunostimulants and act as delivery systems based on their mechanisms of action [148]. Immunostimulants, including PAMPs, DAMPs, and other small-molecule agonists, trigger danger signals that activate pattern recognition receptors (PRRs) on APCs. Additionally, PRR activation upregulates cytokine and co-stimulatory molecule expression, leading to increased co-stimulatory signaling, which together enhances the presentation of antigens on MHC molecules. Delivery systems, such as LNPs, PLGA nanoparticles, and self-loaded protein nanoparticles, function by facilitating antigen presentation on MHC molecules.

mRNA vaccines exhibit self-adjuvant properties [149]. Although the by-product dsRNA may cause side effects, whether low doses of dsRNA can be used as an adjuvant to enhance vaccine efficacy needs to be explored. Conventional adjuvant strategies are applicable to mRNA vaccines. New LNP components integrating TLR7/8 agonist activity enhance the immunogenicity and delivery efficiency of SARS-CoV-2 mRNA vaccines [150]. Adding ATP to LNP significantly improves mRNA expression and enhances delivery efficiency [151]. All-trans retinoic acid (ATRA) [152] integration in LNPs activates DCs in lymph nodes, promoting T cell response and tumor immunity. Using Mn ions to activate STING promotes APC maturation and mRNA expression [153], while encoding mRNA for the activation of AP-1cGASAgonists further enhances immunogenicity [154].

Similarly, mRNA can directly encode immune stimulatory molecules, achieving a built-in adjuvant effect. These self-generated adjuvants include but are not limited to chemokines and complements. Introducing MHC class I trafficking signals or lysosomal/endosomal localization signals assists the process of MHC class I or class II epitope presentation by DCs [155]. Gene fusion of tetanus toxoid fragments with antigens has been widely used in both DNA and mRNA vaccine design [156], which activate helper T cells. Fusion with C3 complement also improved the efficacy of mRNA vaccines [157]. Chemokines are promising molecular adjuvants in mRNA vaccines. Delivering antigens together with the interaction between natural ligands such as chemokines and specific receptors on the surface of APCs enhances the antigenicity of mRNA vaccines [31,158].

The integrated application of these strategies not only optimizes the adjuvant properties of mRNA vaccines but also greatly enhances the immune response and therapeutic potential of vaccines through innovative delivery systems and immune-activation mechanisms.

### 2.7. Security Issues

LNPs, the footstone of RNA therapeutics, face several challenges. mRNA vaccines can induce strong immune responses. However, the use of Mrna-lnp vaccines is associated with dose-limiting systemic inflammatory responses in humans. N1mΨ-UTP amplifies the release of IL-1-related broad-spectrum pro-inflammatory cytokines [159]. The dynamics of DC subsets and NKT-like cells correlate with the reactogenicity of the SARS-CoV-2 mRNA vaccine [160].

#### 2.7.1. Cationic Lipid Oxidation Injuries in the Safety of LNPs

The widespread application of LNPs is limited by their inherent instability and stringent ultra-cold storage and transportation requirements. The cationic lipid components, crucial for mRNA delivery, are prone to oxidation, leading to toxicity concerns. Although Tris-buffer has been shown to partially mitigate the safety and efficacy issues associated with the oxidation of ionizable cationic lipids, the development of thermostable LNPs is another research interest [161]. iLAND is a high-temperature-resistant siRNA drug delivery system for the clinical treatment of hyperlipidemia [162]. SiSaf’s silicon-stabilized hybrid lipid nanoparticles (sshLNPs) are another answer, which significantly enhance the thermal stability and transfection efficiency of siRNA while reducing the immune response and accumulation in vivo [163]. Recently, AstraZeneca showed an innovative inhalable LNP-mRNA dry powder formulation that achieves stability during room-temperature drying by optimizing the formulation, opening up a new pathway for mRNA pulmonary delivery and marking an important advance in LNP technology for drug delivery [164].

#### 2.7.2. Anti-PEG Antibody: Potential Barriers for mRNA Application

Although PEG are widely used in drug delivery due to their enhanced drug solubility and reduced immunogenicity, new evidence revealed that PEG can induce the formation of anti-PEG antibodies, causing accelerated blood clearance and affecting drug efficacy, especially for mRNA vaccines based on LNPs [165]. Studies have demonstrated that LNP-mediated post-injection production of anti-PEG antibodies is dose-dependent, particularly anti-PEG IgG, which requires multiple injections to appear, indicating that PEG can induce immune memory [166]. Continuous injection of PEG-LNPs leads to accelerated clearance by the reticuloendothelial system, which may trigger hypersensitivity or infusion reactions. The increased exposure to PEG also promoted the demand for alternative LNP formulations without PEG components. To address this issue, some studies use polyinosine (pSar) lipids as alternatives to PEG lipids, and the complete replacement of the PEG lipid with a pSar lipid can increase or maintain mRNA delivery efficiency and exhibit similar safety profiles in vivo [167].

## 3. Conclusions

The rapid advancements in mRNA technology hold immense promise for the future of medicine. This article reviewed the latest progress in key technologies such as codon optimization, IVT, and LNP delivery systems, providing a comprehensive perspective for understanding this cutting-edge field.

## Figures and Tables

**Figure 1 vaccines-12-00873-f001:**
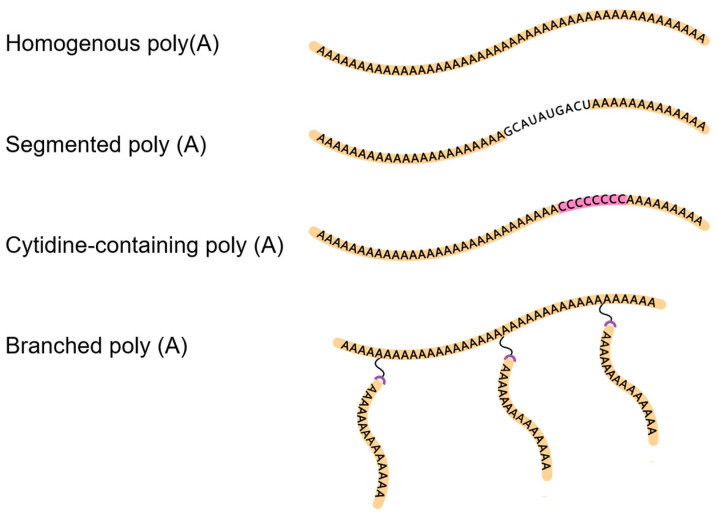
Different poly(A) tails.

**Figure 2 vaccines-12-00873-f002:**
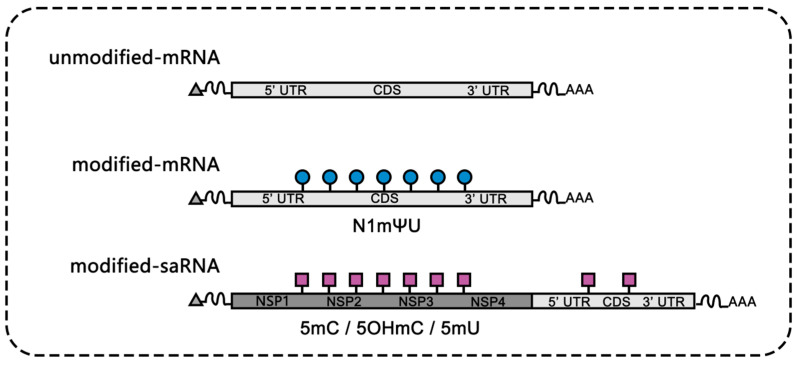
Nucleotide modifications in mRNA vaccines.

**Figure 3 vaccines-12-00873-f003:**
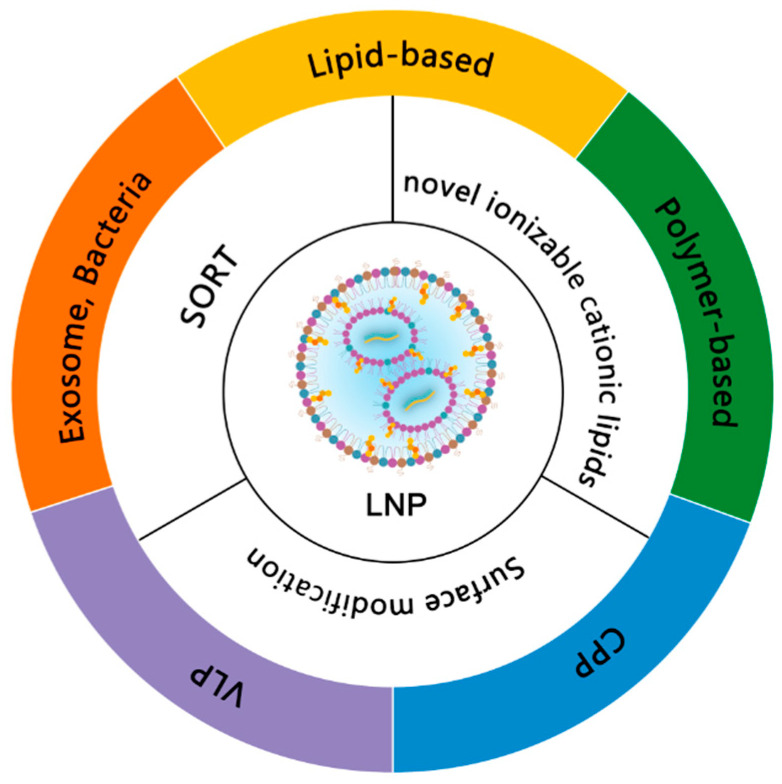
Delivery systems for mRNA.

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
