# Peer review of "Recent Advancements in mRNA Vaccines: From Target Selection to Delivery Systems"

_vaccines, 2024, doi:10.3390/vaccines12080873_

Round 1
Reviewer 1 Report
Comments and Suggestions for Authors
Since the COVID-19 pandemic and the success of the two mRNA vaccines, several researchers and ample groups have been working on developing novel mRNA vaccines and therapeutics. Wu et al. have submitted the review entitled "Recent Advancements in mRNA Vaccines: From Target Selection to Delivery Systems.".
The authors have understood most aspects of the mRNA platform, including its design, construction, and formulation.
The literature collection was extensive, highlighting important aspects.
At my side, I do not have any comments; however, it would be good to have the maximum number of illustrations wherever possible, for example, poly-A repeat sequences and others.
Reviewer 2 Report
Comments and Suggestions for Authors
The authors are comprehensively reviewing the current state-of-the-art in the field of RNA vaccines. They first provide an overview of the field of therapeutic RNA application, before they dig deeper into RNA vaccine design, which includes antigen and RNA engineering. They review improved RNA synthesis methods using modified nucleotides, cap analogs and measures to remove double stranded RNA contaminants. The last chapters of the paper discuss RNA delivery, the use of adjuvants and safety aspects of RNA vaccines.
General comment:
Overall, the paper provides a very good overview of the current status of mRNA vaccine development and I recommend publication upon some revisions to the text. Unfortunately, it is quite exhausting to read, as some passages are difficult to understand. It is often not entirely clear what the authors mean exactly, and some expressions are used imprecisely (see specific comments below). In addition, the sentence structure is sometimes complicated and there are many abbreviations that have not been defined. The authors might also want to revisit if all abbreviated terms are required and thereby reduce the overall number of abbreviations. Although I am not a native English speaker myself, I felt that the paper as a whole would benefit from a critical review in terms of sentence structure and word choice, as well as a general proofreading by a native speaker if not done yet.
Specific comments:
lines 25/26: please revisit placement of commas
Lines 45/46: The following sentence is misleading: “mRNA vaccines only contain mRNA without the risk of genome integration”. It somehow implies that other mRNAs have the risk of integration. What is meant here? Probably that opposed to DNA, mRNA avoids genomic integration?
Lines 51-53: Another benefit of RNA that should be mentioned is that the chemistry of mRNA is independent of the encoded antigens. Therefore, vaccine production does not require adaptation when changing antigens.
Line 65: Why does the multi-step IVT and LNP formulation limit the scale of production? Could the authors explain this a bit more? I think this dramatically raises costs of large batches, but does it really limit the batch size by itself?
Lines 68/69: Could the author's be more precise what they mean? Is the risk of myocarditis related to the use of mRNA, or related to the immune response to encoded SARS-CoV-2 antigens?
Lines 77/78: saRNA should be explained in greater detail, since some readers might not be familiar with the different RNA-formats.
Lines 93/94: What is a CCL11-fusion?
Lines 95-101, and lines 113-115: The authors should mention at which stage all these developments are. Are we talking about findings in mice, NHP or other animals, clinical testing?
line 123: Abbreviation “bsAb” is not defined
Line 125: What does the term “molecules” stand for? mRNA, or also others? GSDMD and TRAIL, which are mentioned as examples, are proteins. Therefore I suggest to use a term like “mRNA encoded therapeutic proteins” or similar
Line 141: Which protein is encoded by mRNA-3927?
Line 148: Do the authors mean a transient or permanent reprograming of the cells?
Line 156: The selection of the proper antigen is crucial for any vaccine, not only mRNA.
Lines 164-166: This sentence is not clear to me. mRNA leads almost inevitably to overexpression, what do the authors mean here? Should translation levels be controlled? Maybe "avoid" is not the right word (“assess the risk”?)... or syntax needs to be improved.
Lines 170-172: The sentence reads: “This aims to make the expressed S protein more stable in the conformation before binding to host cell receptors, thereby producing neutralizing antibodies that block its invasion into host cells.” Do I understand correctly, that the S protein itself is prone to invade cells unless modified in the described manner?
Lines 181, 189: Does “epitope selection” and “multi-epitope design” refer to B- or T-cell epitopes, or both? Overall in this paragraph about epitope optimization, the authors should state clearer whether “epitope” stands for a peptide recognized by antibodies or T cell receptors, or a whole protein/domain. For instance, the cited study by Arevalo et al 2022 used full-length HA, not epitopes.
Line 234: What do the authors mean by “overshadowed”?
Line 259/260: Please delete the title of reference 59 from the text.
Line 286/287: The authors should also state that the segmented polyA mentioned above cannot be generated by enzymatic poly-adenylation.
Line 311: Why do the authors mention explicitly nsP1? To my knowledge, it is not fully understood why 1mY modification is incompatible with saRNA. It could also be due to nsP4.
Lines 328 and following: The paragraph about cap analogs would benefit from a brief recapitulation of different cap structures in mammalian cells, before going into more details.
Line 335: Since you are talking about cap-2 in the sentence right before, please mention that SARS-CoV-2 genomic RNA is bearing a cap-1!
Line 338/339: More precisely, in vitro conversion of cap-0 to cap-1 is done by post-transcriptional methylation. And more precisely, commercially available ARCA cap analogs introduce a cap-0, unless enzymatically methylated in a second step.
Line 343: It should be mentioned that the superior capping efficiency of CleanCap is related to the fact that it does not compete with GTP duting IVT at the initiation of transcripts.
Line 345: “a vital auxiliary component” should be replaced by “an essential component”
Line 361 and following: In the paragraph about dsRNA and dsRNA removal the method of purification by cellulose should be mentionend (Baiersdörfer et al., 2019. PMID: 30933724)
Line 383. Chapter about delivery: I suggest to restructure the chapter. Please consider to start the chapter with LNPs, since they were used with the approved mRNA vaccines and are the most widely used in mRNA literature. Furthermore, I suggest to move “Lipids for advanced LNPs” up and put it directly after the LNP.
Line 476: Please mention which health authorities have approved the Siwei vaccine.
Line 512: What do the authors mean by “the bond in the tail…”?
Line 547: The sentence “The in vivo cart-coupled antibody achieves targeted delivery.” is not understandable for people outside the CART field. Please consider rephrasing.
Comments on the Quality of English LanguageSee my general comment above
Author Response
Response to Reviewer 2 Comments
- Summary
Thank you very much for taking the time to review our manuscript. Please find the detailed responses below and the corresponding revisions/corrections highlighted/in track changes in the re-submitted files.
- Point-by-point response to Comments and Suggestions for Authors
Comments 1: Overall, the paper provides a very good overview of the current status of mRNA vaccine development and I recommend publication upon some revisions to the text. Unfortunately, it is quite exhausting to read, as some passages are difficult to understand. It is often not entirely clear what the authors mean exactly, and some expressions are used imprecisely (see specific comments below). In addition, the sentence structure is sometimes complicated and there are many abbreviations that have not been defined. The authors might also want to revisit if all abbreviated terms are required and thereby reduce the overall number of abbreviations. Although I am not a native English speaker myself, I felt that the paper as a whole would benefit from a critical review in terms of sentence structure and word choice, as well as a general proofreading by a native speaker if not done yet.
Response 1: We thank for your critical and constructive comments, which helped us to improve our manuscript. We have polished our manuscript according to the detailed comments.
Comments 2: lines 25/26: please revisit placement of commas
Response 2: We have corrected the placement of commas in the revised version as follows:
“Through large-scale vaccination, humans completely eradicated smallpox in the 1980s, the deadliest infectious disease that had caused millions of deaths.”
Comments 3: Lines 45/46: The following sentence is misleading: “mRNA vaccines only contain mRNA without the risk of genome integration”. It somehow implies that other mRNAs have the risk of integration. What is meant here? Probably that opposed to DNA, mRNA avoids genomic integration?
Response 3: We apologize for our imprecise and misleading description and have provided a more accurate sentence in the revised version as follows:
“Unlikw DNA, mRNA vaccines avoid the risk of genomic integration.”
Comments 4: Lines 51-53: Another benefit of RNA that should be mentioned is that the chemistry of mRNA is independent of the encoded antigens. Therefore, vaccine production does not require adaptation when changing antigens.
Response 4: We appreciate your insightful suggestion. As suggested, we have discussed this point in the revised version as follows:
“The chemical structure of mRNA itself is independent of the antigen it encodes, and this characteristic endows mRNA vaccine production with modularity and flexibility. Therefore, mRNA vaccine production does not require adaptation when changing antigens.”
Comments 5: Line 65: Why does the multi-step IVT and LNP formulation limit the scale of production? Could the authors explain this a bit more? I think this dramatically raises costs of large batches, but does it really limit the batch size by itself?
Response 5: We agree with that the large-scale raises costs of large batches and have revised it as follows:
“The large-scale of LNP formulation is costly and time-consuming. Encapsulating RNA in LNPs requires highly controlled mixing of RNA and lipids in the microfluidic chips, and it is hard to scale up to high-throughput production.”
Comments 6: Lines 68/69: Could the author's be more precise what they mean? Is the risk of myocarditis related to the use of mRNA, or related to the immune response to encoded SARS-CoV-2 antigens?
Response 6: We agree with that the risk of myocarditis is related to the immune response to encoded SARS-CoV-2 antigens, and have changed it as follows:
“Thirdly, mRNA vaccines may have strong side effects, such as fever and headache. The COVID-19 mRNA vaccine may lead to a T-cell-mediated autoimmune liver disease [1]. Moreover, the COVID-19 mRNA vaccine encoded spike protein is related to the risk of myocarditis, which is more pronounced in young males [2, 3].”
Comments 7: Lines 77/78: saRNA should be explained in greater detail, since some readers might not be familiar with the different RNA-formats.
Response 7: As suggested, we have added a brief description of saRNA in the revised version as follows:
“saRNA is a type of mRNA molecule that has been engineered to replicate itself within host cells. saRNA utilizes the four non structural proteins (NSP1-4) encoding the replicase of the alphavirus and the subgenomic promoter to achieve self amplification, allowing for enhanced protein expression and a stronger immune response [4].”
Comments 8: Lines 93/94: What is a CCL11-fusion?
Response 8: We apologize for our imprecise description and have provided a more accurate description in the revised version as follows:
“Our work showed that mRNA vaccines encoding HPV-E6E7 fused with chemokine CCL11 treatment achieved durable complete remission in tumor animal models.”
Comments 9: Lines 95-101, and lines 113-115: The authors should mention at which stage all these developments are. Are we talking about findings in mice, NHP or other animals, clinical testing?
Response 9: We apologize for the missing of key information and have introduced the background of these results in the revised version.
Comments 10: line 123: Abbreviation “bsAb” is not defined
Response 10: We have substitute the abbreviation “bsAb” for bispecific antibody in the revised version.
Comments 11: Line 125: What does the term “molecules” stand for? mRNA, or also others? GSDMD and TRAIL, which are mentioned as examples, are proteins. Therefore I suggest to use a term like “mRNA encoded therapeutic proteins” or similar
Response 11: We thank for your constructive comments. As suggested, we have replaced the “molecules” with a more specific term “mRNA encoded therapeutic proteins”.
Comments 12: Line 141: Which protein is encoded by mRNA-3927?
Response 12: We thank for your question, and have described the translational product of mRNA-3927 in the revised version as follows:
“mRNA-3927 encodes PCCA and PCCB subunit proteins to restore functional propionyl-coenzyme carboxylase.”
Comments 13: Line 148: Do the authors mean a transient or permanent reprograming of the cells?
Response 13: We apologize for our imprecise description and have revised it in the revised version as follows:
“Transgenic overexpression of hepatocyte growth factor can increase insulin production and β cell proliferation [5]. Similarly, the transcription factors Pdx1 and MafA can reprogram pancreatic alpha cells into functional, insulin-producing β cells [6]. mRNA-LNPs targeting the pancreas hold immense potential for the treatment of type 1 diabetes by delivering mRNA encoding these proteins [7].”
Comments 14: Line 156: The selection of the proper antigen is crucial for any vaccine, not only mRNA.
Response 14: We thank for your constructive comments and have rephrased it in the revised version as follows:
“The selection of antigens is crucial for the success of vaccines”
Comments 15: Lines 164-166: This sentence is not clear to me. mRNA leads almost inevitably to overexpression, what do the authors mean here? Should translation levels be controlled? Maybe "avoid" is not the right word (“assess the risk”?)... or syntax needs to be improved.
Response 15: Overexpression of antigens is inevitable, and some antigens, such as HPV E6 protein, are oncogenic. Therefore, in the design process of tumor vaccines, it is necessary to mutate relevant sites to avoid oncogenicity. We apologize for our misleading description, and rephrased it in the revised version as follows:
“After selecting the antigen, it is essential to conduct a detailed investigation and analysis of its biological function. On one hand, it is necessary to avoid the risk of overexpressing the natural oncogenic antigen. For instance, the HPV E6 protein has the ability to degrade P53 and is considered a proto-oncoprotein; thus, mutations in the corresponding binding sites are required in vaccine design [8]. On the other hand, the appropriate conformation helps for the effective immune response. In the design of COVID-19 vaccines, a double proline mutation in the S protein is used to stabilize its pre-fusion conformation [9]. This aims to make the expressed S protein more stable in the conformation before binding to host cell receptors, thereby producing neutralizing antibodies that block its invasion into host cells.”
Comments 16: Lines 170-172: The sentence reads: “This aims to make the expressed S protein more stable in the conformation before binding to host cell receptors, thereby producing neutralizing antibodies that block its invasion into host cells.” Do I understand correctly, that the S protein itself is prone to invade cells unless modified in the described manner?
Response 16: We thank for your question. We think that the neutralizing antibodies against the pre-fusion conformation of S protein are more effective in blocking host invasion.
Comments 17: Lines 181, 189: Does “epitope selection” and “multi-epitope design” refer to B- or T-cell epitopes, or both? Overall in this paragraph about epitope optimization, the authors should state clearer whether “epitope” stands for a peptide recognized by antibodies or T cell receptors, or a whole protein/domain. For instance, the cited study by Arevalo et al 2022 used full-length HA, not epitopes.
Response 17: We agree with that the multivalent influenza mRNA vaccine used full-length HA, not epitopes. Antigenic epitope is the basic unit that triggers cellular or humoral immune responses. We apologize for our imprecise statement and have rephrased this part as follows:
“Multi-epitope vaccine is a promising strategy against tumors and viral infections Compared to classical vaccines and single-epitope vaccines, multi epitope design aims to enhance the immunogenicity of vaccines and cover immune responses in different populations. Multi-epitope design concludes â… ) multiple B or T epitopes, â…¡) combined B and T cell epitopes, â…¢) multiple epitopes from different tumor or virus antigens [10]. The monkeypox mRNA vaccines MPXVac-097 combined 5 MPXV antigens that have been identified targets of neutralizing antibodies induces broad-spectrum neutralizing antibodies and specific T cell responses by tandemly linking five antigens on a single mRNA molecule and provides protection against live virus challenge in animal models [11]. Multi epitope vaccines consisting of B and T cell epitopes have the potential to trigger innate, adaptive, and humoral responses [12]. mRNA vaccines encoding HLA-EPs (T epitopes) and the receptor-binding domain of the SARS-CoV-2 (neutralizing antibodies) produces optimal protection against SARS-CoV-2 in nonhuman primates [13]. Multivalent HPV DNA/mRNA vaccines encoding HPV16/18 E6/E7 epitopes induced significant antigen-specific cellular immune responses in mice CIN3 patients [14, 15].”
Comments 18: Line 234: What do the authors mean by “overshadowed”?
Response 18: We apologize for our confusing statement and have changed the sentence in the revised version as follows:
“The 5' and 3' untranslated regions (UTRs) of mRNA, which are less likely to attract attention compared to the coding sequence, play a critical role in protein expression.”
Comments 19: Line 259/260: Please delete the title of reference 59 from the text.
Response 19: We apologize for this obvious mistake and have fixed the error.
Comments 20: Line 286/287: The authors should also state that the segmented polyA mentioned above cannot be generated by enzymatic poly-adenylation.
Response 20: We agree with that only Co-IVT can generate segmented polyA and have discussed this issue in the revised version as follows:
“Moreover, only co-transcriptional synthesis can generate segmented polyA.”
Comments 21: Line 311: Why do the authors mention explicitly nsP1? To my knowledge, it is not fully understood why 1mY modification is incompatible with saRNA. It could also be due to nsP4.
Response 21: We agree with that NSP4 (RdRp) is most likely to be incompatible with N1mΨ-UTP and have corrected it in the revised version.
Comments 22: Lines 328 and following: The paragraph about cap analogs would benefit from a brief recapitulation of different cap structures in mammalian cells, before going into more details.
Response 22: As suggested, we have introduced the three different cap structures in the revised version as follows:
“Cap-0 is a N7-methyl guanosine connected to the 5′ nucleotide through a 5′ to 5′ triphosphate linkage. The Cap-0 structure is essential for efficient translation of the mRNA. An additional methylation on the 2′O position of the initiating nucleotide generates Cap-1. Cap1 evades the recognition by the host cell innate immune system. Notably, viruses like SARS-CoV-2 encode 2'-O-MTase to strategically modify their RNA caps, facilitating immune evasion and promoting stable translation of viral RNA [16]. Cap2 is formed through a slow continuous conversion of mRNAs from Cap1 to Cap2 in the cytosol, with an additional methylation on the 2′O position of the second nucleotide. Recent study showed that Cap2 is enriched on long-lived mRNAs, and the methylation of Cap1 to Cap2 markedly reduces the ability of RNAs to bind to and activate RIG-I [17].“
Comments 23: Line 335: Since you are talking about cap-2 in the sentence right before, please mention that SARS-CoV-2 genomic RNA is bearing a cap-1!
Response 23: We apologize for our confusing description and have rephrased it in the revised version as follows:
“Cap-0 is a N7-methyl guanosine connected to the 5′ nucleotide through a 5′ to 5′ triphosphate linkage. The Cap-0 structure is essential for efficient translation of the mRNA. An additional methylation on the 2′O position of the initiating nucleotide generates Cap-1. Cap1 evades the recognition by the host cell innate immune system. Notably, viruses like SARS-CoV-2 encode 2'-O-MTase to strategically modify their RNA caps, facilitating immune evasion and promoting stable translation of viral RNA [16]. Cap2 is formed through a slow continuous conversion of mRNAs from Cap1 to Cap2 in the cytosol, with an additional methylation on the 2′O position of the second nucleotide. Recent study showed that Cap2 is enriched on long-lived mRNAs, and the methylation of Cap1 to Cap2 markedly reduces the ability of RNAs to bind to and activate RIG-I [17]. “
Comments 24: Line 338/339: More precisely, in vitro conversion of cap-0 to cap-1 is done by post-transcriptional methylation. And more precisely, commercially available ARCA cap analogs introduce a cap-0, unless enzymatically methylated in a second step.
Response 24: We appreciate your insightful comments. As suggested, we have pointed this in the revised version.
Comments 25: Line 343: It should be mentioned that the superior capping efficiency of CleanCap is related to the fact that it does not compete with GTP duting IVT at the initiation of transcripts.
Response 25: We appreciate your critical suggestion and have pointed this in the revised version.
Comments 26: Line 345: “a vital auxiliary component” should be replaced by “an essential component”
Response 26: As suggested, we have substitute “an essential component” for “a vital auxiliary component” in the revised version.
Comments 27: Line 361 and following: In the paragraph about dsRNA and dsRNA removal the method of purification by cellulose should be mentionend (Baiersdörfer et al., 2019. PMID: 30933724)
Response 27: We appreciate your critical comments. As suggested, we have pointed this method in the revised version as follows:
“Purification by cellulose can remove 90% of the dsRNA contaminants, offering an effective, reliable, and safe method [18].”
Comments 28: Line 383. Chapter about delivery: I suggest to restructure the chapter. Please consider to start the chapter with LNPs, since they were used with the approved mRNA vaccines and are the most widely used in mRNA literature. Furthermore, I suggest to move “Lipids for advanced LNPs” up and put it directly after the LNP.
Response 28: We appreciate your constructive comments. As suggested, we have rearranged the chapter about delivery in the revised version.
Comments 29: Line 476: Please mention which health authorities have approved the Siwei vaccine.
Response 29: We thank for your critical comments and have rephrased it in the revised version as follows:
“The COVID-19 vaccine SW0123 from Stemirna Therapeutics adopts this delivery strategy [19], and the clinical trial application for the LPP-based mRNA vaccine has been accepted by the CDE (China's National Medical Products Administration) and the clinical trials have been conducted. In addition, the LPP-based mRNA vaccine has been granted an Emergency Use Authorization by Laos.”
Comments 30: Line 512: What do the authors mean by “the bond in the tail…”?
Response 30: We apologize for our confusing description and have provided a more accurate description of this issue in the revised version as follows:
“Chemical bonds at the tail of ionizable cationic lipids influences the targeting of LNPs.”
Comments 31: Line 547: The sentence “The in vivo cart-coupled antibody achieves targeted delivery.” is not understandable for people outside the CART field. Please consider rephrasing.
Response 31: We appreciate your critical suggestion and have rephrased this sentence for easy reading in the revised version as follows:
“In vivo CART refers to the production of CART cells in vivo, without the need for ex vivo modification and transfusion. By coupling CD5 antibodies on the surface of LNPs (Ab-LNPs), these particles are used to carry CAR mRNA and target T cells for In vivo CART production [20].”
- T. Boettler et al., SARS-CoV-2 vaccination can elicit a CD8 T-cell dominant hepatitis. J Hepatol 77, 653-659 (2022).
- L. M. Yonker et al., Circulating Spike Protein Detected in Post-COVID-19 mRNA Vaccine Myocarditis. Circulation 147, 867-876 (2023).
- M. Fan et al., Risk of carditis after three doses of vaccination with mRNA (BNT162b2) or inactivated (CoronaVac) covid-19 vaccination: a self-controlled cases series and a case-control study. Lancet Reg Health West Pac 35, 100745 (2023).
- A. K. Blakney, S. Ip, A. J. Geall, An Update on Self-Amplifying mRNA Vaccine Development. Vaccines (Basel) 9, (2021).
- A. Garcia-Ocana et al., Hepatocyte growth factor overexpression in the islet of transgenic mice increases beta cell proliferation, enhances islet mass, and induces mild hypoglycemia. J Biol Chem 275, 1226-1232 (2000).
- X. Xiao et al., Endogenous Reprogramming of Alpha Cells into Beta Cells, Induced by Viral Gene Therapy, Reverses Autoimmune Diabetes. Cell Stem Cell 22, 78-90.e74 (2018).
- J. R. Melamed et al., Ionizable lipid nanoparticles deliver mRNA to pancreatic β cells via macrophage-mediated gene transfer. Sci Adv 9, eade1444 (2023).
- A. A. McBride, Human papillomaviruses: diversity, infection and host interactions. Nat Rev Microbiol 20, 95-108 (2022).
- C. L. Hsieh et al., Structure-based design of prefusion-stabilized SARS-CoV-2 spikes. Science 369, 1501-1505 (2020).
- L. Zhang, Multi-epitope vaccines: a promising strategy against tumors and viral infections. Cell Mol Immunol 15, 182-184 (2018).
- Z. Fang et al., Polyvalent mRNA vaccination elicited potent immune response to monkeypox virus surface antigens. Cell Res 33, 407-410 (2023).
- S. Aziz et al., Contriving multi-epitope vaccine ensemble for monkeypox disease using an immunoinformatics approach. Front Immunol 13, 1004804 (2022).
- W. Tai et al., An mRNA-based T-cell-inducing antigen strengthens COVID-19 vaccine against SARS-CoV-2 variants. Nat Commun 14, 2962 (2023).
- T. J. Kim et al., Clearance of persistent HPV infection and cervical lesion by therapeutic DNA vaccine in CIN3 patients. Nat Commun 5, 5317 (2014).
- J. Wang et al., Development of an mRNA-based therapeutic vaccine mHTV-03E2 for high-risk HPV-related malignancies. Mol Ther 32, 2340-2356 (2024).
- A. Paramasivam, RNA 2'-O-methylation modification and its implication in COVID-19 immunity. Cell Death Discov 6, 118 (2020).
- V. Despic, S. R. Jaffrey, mRNA ageing shapes the Cap2 methylome in mammalian mRNA. Nature 614, 358-366 (2023).
- M. Baiersdorfer et al., A Facile Method for the Removal of dsRNA Contaminant from In Vitro-Transcribed mRNA. Mol Ther Nucleic Acids 15, 26-35 (2019).
- R. Yang et al., A core-shell structured COVID-19 mRNA vaccine with favorable biodistribution pattern and promising immunity. Signal Transduct Target Ther 6, 213 (2021).
- J. G. Rurik et al., CAR T cells produced in vivo to treat cardiac injury. Science 375, 91-96 (2022).
Reviewer 3 Report
Comments and Suggestions for Authors
The authors have submitted the manuscript titled "Recent advancements in mRNA Vaccines: From Target Selection to Delivery Systems". In this, the authors summarize the recent advancements in mRNA vaccines including target selection and delivery systems.
Overall it is a very well-written manuscript. The authors seem to have thoroughly analyzed the literature.
I only have a minor comment. The authors should add a section on the drawbacks or disadvantages of the current mRNA vaccines and future directions for research in this area. This will make the review complete and all-encompassing.
Comments on the Quality of English LanguageThere are a few (2 or 3) grammatical mistakes in the manuscript. These should be corrected. Other that that, the English language is fine.
Author Response
Response to Reviewer 3 Comments
- Summary
Thank you very much for taking the time to review our manuscript. Please find the detailed responses below and the corresponding revisions/corrections highlighted/in track changes in the re-submitted files.
- Point-by-point response to Comments and Suggestions for Authors
Comments 1: The authors have submitted the manuscript titled "Recent advancements in mRNA Vaccines: From Target Selection to Delivery Systems". In this, the authors summarize the recent advancements in mRNA vaccines including target selection and delivery systems.
Overall it is a very well-written manuscript. The authors seem to have thoroughly analyzed the literature. I only have a minor comment. The authors should add a section on the drawbacks or disadvantages of the current mRNA vaccines and future directions for research in this area. This will make the review complete and all-encompassing.
Response 1: We appreciate your constructive comments. As suggested, we have discussed the disadvantages and side effects of mRNA-LNP more detailedly in the revised manuscript.
Comments 2: Comments on the Quality of English Language
There are a few (2 or 3) grammatical mistakes in the manuscript. These should be corrected. Other that that, the English language is fine.
Response 2: We appreciate your criticism comments, and have carefully checked and revised our manuscript.
Reviewer 4 Report
Comments and Suggestions for Authors
While the authors did a great job discussing essential parts of mRNA vaccines, there remains essential aspects that are missing in the current manuscript, which I feel the authors should discuss:
(1) Immunological mechanisms for mRNA vaccines to induce immunity. Here are several excellent research articles that may be discussed: 10.1016/j.immuni.2020.11.009, 10.1038/s41586-021-03791-x, 10.1038/s41590-022-01163-9
(2) 2.7. Safety of LNPs as vaccines. The authors did not discuss the most significant safety concern for mRNA LNPs as vaccines, which should be reactogenicity and cytokine release-like side reactions. Here are some examples that discuss this topic in detail and should be great references: 10.1038/s41590-022-01160-y, 10.1016/j.xcrm.2022.100631
Typo in Abstract: Line 17. ...for further improvement *of* the immunogenicity...
The discussions on solid lipid nanoparticles and nanostructured lipid carriers are misleading. The LNPs used in clinically approved mRNA vaccines should or can also be considered solid lipid nanoparticles or nanostructured lipid carriers. At physiological pH, LNPs possess a solid (amorphous) core appearing electron-dense under cryo-TEM, and there are many literature reports debating their internal structures. But what's certain is that they are solid, while they have internal structures. I would say that given the current landscape of LNPs (I'm referring to those made with ionizable lipids), there is no point to distinguish clearly what is an LNP, what is an SLN, or what is an NLC. I suggest the authors make clear of this point, and reference 93 cited seems to be completely irrelevant to this point. For references, some excellent papers which may be useful:
10.1021/acsnano.8b01516, 10.1038/s41467-022-33157-4, 10.1021/jp303267y
Author Response
Response to Reviewer 4 Comments
- Summary
Thank you very much for taking the time to review our manuscript. Please find the detailed responses below and the corresponding revisions/corrections highlighted/in track changes in the re-submitted files.
- Point-by-point response to Comments and Suggestions for Authors
Comments 1: While the authors did a great job discussing essential parts of mRNA vaccines, there remains essential aspects that are missing in the current manuscript, which I feel the authors should discuss: (1) Immunological mechanisms for mRNA vaccines to induce immunity. Here are several excellent research articles that may be discussed: 10.1016/j.immuni.2020.11.009, 10.1038/s41586-021-03791-x, 10.1038/s41590-022-01163-9
Response 1: We thank for your constructive suggestions, and have discussed the immunological mechanisms for mRNA vaccines to induce immunity in the revised version of the manuscript as follows:
“The SARS-CoV-2 mRNA vaccines promote the germinal center responses and produce effective neutralizing antibodies, as well as antigen-specific polyfunctional CD4 and CD8 T cells [1, 2]. Moreover, second vaccination stimulated a notably enhanced innate immune response, which was concurrent with enhanced serum IFN-γ levels. Natural killer cells and CD8+ T cells in the draining lymph nodes are the major producers of this circulating IFN-γ [3].”
Comments 2: (2) 2.7. Safety of LNPs as vaccines. The authors did not discuss the most significant safety concern for mRNA LNPs as vaccines, which should be reactogenicity and cytokine release-like side reactions. Here are some examples that discuss this topic in detail and should be great references: 10.1038/s41590-022-01160-y, 10.1016/j.xcrm.2022.100631
Response 2: We appreciate your constructive comments and have discussed the reactogenicity and cytokine release-like side reactions in the revised version of the manuscript as follows:
“mRNA vaccines can induce strong immune responses. However, the use of Mrna-lnp vaccines is associated with dose-limiting systemic inflammatory responses in humans. N1mΨ-UTP amplifies the release of IL-1 related broad-spectrum pro-inflammatory cytokines [4]. The dynamics of DC subsets and NKT-like cells correlate with the reactogenicity of SARS-CoV-2 mRNA vaccine [5].”
Comments 3: Typo in Abstract: Line 17. ...for further improvement *of* the immunogenicity...
Response 3: Response: We thank for your comments, and have corrected it in the revised manuscript.
Comments 4: The discussions on solid lipid nanoparticles and nanostructured lipid carriers are misleading. The LNPs used in clinically approved mRNA vaccines should or can also be considered solid lipid nanoparticles or nanostructured lipid carriers. At physiological pH, LNPs possess a solid (amorphous) core appearing electron-dense under cryo-TEM, and there are many literature reports debating their internal structures. But what's certain is that they are solid, while they have internal structures. I would say that given the current landscape of LNPs (I'm referring to those made with ionizable lipids), there is no point to distinguish clearly what is an LNP, what is an SLN, or what is an NLC. I suggest the authors make clear of this point, and reference 93 cited seems to be completely irrelevant to this point. For references, some excellent papers which may be useful:
10.1021/acsnano.8b01516, 10.1038/s41467-022-33157-4, 10.1021/jp303267y
Response 4: We apologize for the miscitation and have cited the proper reference in the revised version as follows:
“Liposome is a type of small vesicle with a structure similar to the cell membrane, composed of closed bilayer phospholipid, and is the most classic Lipid based delivery system [6].”
Response 5: We appreciate your constructive comments. After reading the relevant literature, we have a deeper and more accurate understanding of the structure of LNPs, and have revised the relevant content in the latest version based on these references as follows:
“Lipid nanoparticles containing siRNA synthesized through microfluidic mixing exhibit electron dense nanostructure cores[7]. cryo-TEM and X-ray show that siRNA and ionizable lipid form tightly packed bilayer complexes at pH 4, where siRNA is sandwiched between closely apposed monolayers. At physiological pH, these small vesicular structures coalesce to form larger LNP structures with amorphous electron dense cores [8]. Further method based on the multi-laser cylindrical illumination confocal spectroscopy technique quantified the Payload distribution of mRNA-LNP, a commonly referenced benchmark formulation using DLin-MC3 as the ionizable lipid contains mostly 2 mRNAs per loaded LNP with a presence of 40%-80% empty LNPs [9].”
- K. Lederer et al., SARS-CoV-2 mRNA Vaccines Foster Potent Antigen-Specific Germinal Center Responses Associated with Neutralizing Antibody Generation. Immunity 53, 1281-1295 e1285 (2020).
- P. S. Arunachalam et al., Systems vaccinology of the BNT162b2 mRNA vaccine in humans. Nature 596, 410-416 (2021).
- C. Li et al., Mechanisms of innate and adaptive immunity to the Pfizer-BioNTech BNT162b2 vaccine. Nat Immunol 23, 543-555 (2022).
- S. Tahtinen et al., IL-1 and IL-1ra are key regulators of the inflammatory response to RNA vaccines. Nat Immunol 23, 532-542 (2022).
- T. Takano et al., Distinct immune cell dynamics correlate with the immunogenicity and reactogenicity of SARS-CoV-2 mRNA vaccine. Cell Rep Med 3, 100631 (2022).
- T. M. Allen, P. R. Cullis, Liposomal drug delivery systems: from concept to clinical applications. Adv Drug Deliv Rev 65, 36-48 (2013).
- A. K. Leung et al., Lipid Nanoparticles Containing siRNA Synthesized by Microfluidic Mixing Exhibit an Electron-Dense Nanostructured Core. J Phys Chem C Nanomater Interfaces 116, 18440-18450 (2012).
- J. A. Kulkarni et al., On the Formation and Morphology of Lipid Nanoparticles Containing Ionizable Cationic Lipids and siRNA. ACS Nano 12, 4787-4795 (2018).
- S. Li et al., Payload distribution and capacity of mRNA lipid nanoparticles. Nat Commun 13, 5561 (2022).